# Halo Effect and Source Credibility in the Evaluation of Food Products Identified by Third-Party Certified Eco-Labels: Can Information Prevent Biased Inferences?

**DOI:** 10.3390/foods10112512

**Published:** 2021-10-20

**Authors:** Ana Lanero, José-Luis Vázquez, César Sahelices-Pinto

**Affiliations:** Department of Business Economics and Management, Faculty of Economics and Business Sciences, University of León, 24071 León, Spain; jose-luis.vazquez@unileon.es (J.-L.V.); cesar.sahelices@unileon.es (C.S.-P.)

**Keywords:** organic food, third-party certified eco-labels, halo effect, source credibility, quality inferences, price inferences, information

## Abstract

Despite the growing awareness of the need to promote the consumption of organic food, consumers have difficulties in correctly identifying it in the market, making frequent cognitive mistakes in the evaluation of products identified by sustainability labels and claims. This work analyzes the halo effect and the source credibility bias in the interpretation of product attributes based on third-party certified labels. It is hypothesized that, regardless of their specific meaning, official labels lead consumers to infer higher environmental sustainability, quality and price of the product, due to the credibility attributed to the certifying entity. It also examines the extent to which providing the consumer with accurate labeling information helps prevent biased heuristic thinking. An experimental between-subject study was performed with a sample of 412 Spanish business students and data were analyzed using partial least squares. Findings revealed that consumers tend to infer environmental superiority and, consequently, higher quality in products identified by both organic and non-organic certified labels, due to their credibility. Label credibility was also associated with price inferences, to a greater extent than the meaning attributed to the label. Interestingly, providing accurate information did not avoid biased heuristic thinking in product evaluation.

## 1. Introduction

Getting consumers to demand and purchase products from organic farming is a key objective of current far-reaching sustainability policies, appearing as an inseparable complement to the greater involvement of companies and organizations in production processes with less environmental impact [1,2]. Such purpose recognizes the need to establish transparent and reliable systems that aim to verify the use of organic production practices so that markets can operate efficiently, as consumers lack the expertise and time required to obtain accurate information to evaluate them by themselves, either before or after purchase [3,4,5,6]. In this regard, it is known that most of the thinking that takes place in grocery shopping situations occurs heuristically, that is, without enough rational analysis, because consumers often must make several quick decisions in a row and need simple cues to evaluate the attributes of the products with minimal effort [7,8,9]. Thus, heuristics are mental effort-reducing strategies that involve the use of simple decision rules to reach quick and efficient judgements to save time and reduce complexity in personal choice situations [10,11,12].

Therefore, consumers infer beyond given explicit properties to assess the value of unobservable attributes to minimize the risk and uncertainty associated with their choices [13,14,15]. In this regard, third-party certified eco-labels are identification symbols that turn ethical qualities into product characteristics and provide consumers with simple, useful and credible information on complex issues [16,17]. Thus, one of their concrete functions is to act as credibility cues, simplifying consumer decision-making by facilitating the identification of environmentally superior products [3,4,5,16,18]. Furthermore, organic labeling has gained importance as being a tool for building trust in consumers about the quality of the food they consume, as well as to justify its higher price [6,19,20,21,22], which allows reaching market niches less sensitive to environmental sustainability issues compared to other product attributes [16].

The main advantage of heuristic thinking is that it requires much less capacity and effort than rational processing, as it relies on previous learning [23,24,25]. However, its main risk is that it often ignores some of the relevant information and this can lead to biased judgements [11,26,27]. Moreover, the enormous proliferation of eco-labels and other certified sustainability indices has led to a situation of confusion among consumers [3,4,28,29], who tend to have fairly limited knowledge about their precise meaning [30,31,32,33,34,35]. As a result, many of the impressions that consumers form about the sustainability of products are based more on holistic affective evaluations than on qualified reasoning [36,37], which derives from biased inferences about the environmental superiority, quality and price of products [8,32,38].

This paper aims to identify tools that promote effective heuristic thinking in making responsible purchase decisions by analyzing the systematic mistakes committed in the interpretation of attributes of products based on official sustainability indications. In this regard, one of the most frequently studied cognitive biases in the evaluation of sustainable products is the halo effect, which implies the automatic expectation of a higher quality in the case of those products identified as organic [39,40,41,42,43,44,45,46], which in turn is reflected in a higher willingness to pay a premium price [20,21,41,43,47,48,49]. Most intriguingly, such automatic inferences occur even when there is no verifiable evidence of the organic qualities of the product and when consumers do not know exactly what these properties consist of [38,50,51]. Our article expands this line of research by analyzing to what extent it is likely that the halo effect is also generated by certified non-ecological labels, which suggests that confusion between the different dimensions of sustainability [14,32,33] can lead consumers to infer ecological properties and quality expectation in a biased way and thus justify a higher price to be paid for any product identified with a sustainability seal.

Additionally, we contribute to the previous literature with an explanation of this phenomenon based on the so-called source credibility bias, arguing that, in the absence of adequate knowledge, the credibility attributed to a third-party certified label (whether ecological or not) may have a greater degree of influence when compared to the understanding of its meaning on the extent to which a product is perceived as superior at an environmental level and, consequently, on the quality and price inferences associated with it [18,52,53]. In short, an explanation is proposed for the discordance between the use of eco-labels as heuristics of product superiority and the ignorance of their meaning through the concurrence of the source credibility bias (“if it is certified by a third party, it is good”) and the halo effect (“if it is good, it is good in every way”), which in turn help to justify the higher price of the product (“if it is better, it is normal for it to be more expensive”).

In the context of the previous argument, this paper also analyzes the role of information in the prevention of biased heuristic thinking during the assessment of the attributes of responsible products. Starting from the idea that understanding the meaning or third-party certified sustainability labels is an indispensable condition for their adequate use in making purchasing decisions [28,30,35,54,55,56], the conducted research carries out an intervention based on providing information to the consumer to verify its consequences on the credibility and the organic meaning attributed to labels of different dimensions of sustainability, as well as on the resulting inferences of product quality and price.

Relevant practical implications can be derived from the obtained results, either in the case of certifying entities or companies interested in adhering to sustainability labels and efficiently communicating their environmental commitment, as well as for public authorities in charge of designing policies regarding certification of sustainability. Thus, the study of biased heuristic thinking becomes important to identify the cognitive mistakes that systematically occur when interpreting official sustainability labels and to find solutions to them that favor effective heuristic thinking in the achievement of global sustainable consumption goals [8,32,38].

This paper is organized as follows: the next two sections include a review of the main findings in previous literature on the halo effect and source credibility in the evaluation of organic products, as well as the role of information in the prevention of biased heuristic thinking, then proposing a hypothesis model. Thereafter, we present an empirical study to test the model with a Partial Least Squares (PLS) technique. Finally, conclusions and implications are discussed.

## 2. Literature Review

### 2.1. Halo Effect

In consumer decision making, the halo effect occurs when the positive evaluation of an attribute of a product by an individual strongly influences his or her perception of other attributes, so overvaluing the final benefits of the product [41,50,57]. These product associations are generally used as input for consumers to make everyday product-related judgements, such as quality evaluations and value-for-money assessments [58].

Several previous studies show that eco-labels improve the perception of the environmental friendliness of the product [5,59,60], but also other relevant quality attributes, even in settings where there is no reasonable relation between the product label and the evaluated properties [38,50,51]. For instance, some experiments have shown that labeling food as organic improves the way it is perceived in a wide range of judgment perceptions, including sensory assessments such as taste [39,40,43,44,45,46,51,59] and nutritional judgements such as calories and health [38,41,42,51,61,62]. Likewise, it has been found that consumers are usually willing to pay a premium price for food identified as organic, both because of its environmental superiority [21,41,48,51] and because of their own expectations of higher product quality [20,43,45,46,47,49].

Certainly, inferences about the higher quality and price of organic products are prone to effective heuristic thinking in making responsible purchasing decisions when consumers are able to truthfully identify them [8,16,19,21,38]. In this way, eco-labels would fulfill their function of acting as heuristic cues in the identification of better and more expensive products [3,4,5,6,16,18,19,20,21,22]. However, consumers do not seem to know the exact meaning of third-party certified eco-labels [18,30,31,34,35,38]. That is, organic food is considered superior, even when there is no clear knowledge of the characteristics that differentiate it from conventional food [63,64,65,66,67]. Furthermore, consumers do not know how to put into context the meaning of eco-labels regarding other official labels of social or economic sustainability. For instance, some studies show that consumers have rather unclear ideas about the standards and implications of organic and fair-trade labels, so the presence of any of them positively influences the perceived environmental sustainability of the products assessed [32,33]. According to Gruber et al. [14], this can be explained because, in the minds of consumers, sustainability attributes are connected to and contingent on other product attributes, so that individuals evaluate sustainability attributes even when there is no information available about them.

Therefore, whereas the previous literature on the halo effect has focused on the inferences of product superiority associated with environmental labels and claims [5,20,21,38,39,40,41,42,43,44,45,46,47,48,49,50,51,59,60,61,62], it could be a more far-reaching cognitive bias that can be triggered by any certified sustainability seal. That is, in the absence of appropriate knowledge, there is a high risk that consumers mistakenly attribute organic properties and higher quality to products identified by certified non-organic labels, being willing to pay a higher price based on biased premises [14,32,33].

### 2.2. Source Credibility Bias

Going forward with this reasoning, several contributions in the literature show that the influence of eco-labels on purchasing decisions has to do not only with understanding their meaning, but also with the credibility attributed to them [3,6,21,22,34,35,56]. In this sense, many authors conclude that environmental information certified by public authorities and other independent sources is more reliable than that provided by producers or retailers [18,21,68,69,70].

Source credibility is defined as the overall perceptions of individuals regarding the credibility of an information source, rather than the content of the information itself [71]. This occurs when credibility acts as a cognitive shortcut that leads the individual to deem an entity to be legitimate and, therefore, to accept the message of that entity without the need for an exhaustive evaluation of its content [72,73,74].

Precisely, as argued above, one of the main functions of third-party certified eco-labels is to perform as credible symbols that facilitate the identification of environmentally responsible products [3,4,5,16,18]. In fact, because most consumers do not have the technical expertise and knowledge about the requirements that distinguish organic food, experts consider consumer confidence to be a key prerequisite for the emergence of a market for green products [6,18,21,22].

However, it could be argued that while this type of heuristic thinking is effective in generating certainty when purchasing independently certified organic products [8,16,19,21], it also tends to a biased assessment of products when it is erroneously inferred that the credibility of third-party certified labeling legitimizes the superiority of the product in attributes not guaranteed by such certification. This might occur because third-party certifications activate a passive mode of information processing and serve as information cues that enhance perceived legitimacy by consumers [18].

Therefore, since consumers often have difficulties in properly interpreting the meaning of sustainability labels [14,18,30,31,32,33,34,35,38], they are likely to turn to the credibility of the symbol to judge the environmental superiority of the product [6,52,53]. If credibility leads to wrong conclusions about the environmental meaning of the label, biased quality and price inferences might arise. Source credibility bias could thus explain the halo effect generated by non-organic sustainability labels.

### 2.3. The Role of Information in Non-Biased Label Inferences

From the above argument, we can deduce that heuristic inferences are a phenomenon quite ingrained in purchasing decision-making processes based on sustainability labels [7,8,9,38], although they do not ever lead to adequate assessments of product attributes [30,31,32,33,34,35]. Rather, the accuracy of judgements by consumers about the superiority of the product depends on whether they have adequate knowledge to understand the meaning and credibility of the labels [8,9,21,38].

According to the dual models of information processing, improving the knowledge of individuals on a given topic gives them the cognitive ability to make reasoned judgments, thus avoiding cognitive biases and favoring the use of effective heuristic cues [10,12]. In this regard, some contributions in the specialized literature remark that accurate understanding of labels is a prerequisite for label awareness and their correct use in decision-making [28,30,35,54,55,56]. In this way, several studies conclude that consumers who are well informed about the meaning and certification process of eco-labels have a greater capacity to make satisfactory decisions with little effort [65,75,76,77]. This is because the more consumers increase their knowledge on organic food, the more prepared they are to seek specific details about the choice of each product and to use organic claims effectively and correctly as heuristic cues [55,78].

From this perspective, it is acceptable that providing consumers information about the meaning and certification process of different types of third-party sustainability labels should foster the ability of individuals to effectively use them as appropriate heuristic cues in product evaluation [8,16,19,21,38]. Based on this postulate, the next section raises the research hypothesis model.

## 3. Hypotheses

Based on the previous reasoning, this paper advances the research on the halo effect associated with organic labels [5,20,21,38,39,40,41,42,43,44,45,46,47,48,49,50,51,59,60,61,62], analyzing to what extent it can also be generated by other types of non-ecological certifications in uninformed customers, as they do not have adequate knowledge about the meaning of the ones or the others to substantiate their inferences of product quality and price [14,18,30,31,32,33,34,35,38]. Thus, the following hypothesis is proposed:

**Hypothesis** **1 (H1).**
*For uninformed consumers, environmental sustainability inferences have positive effects on quality inferences (H1a) and price inferences (H1b), which apply to both organic and non-organic labels.*


Besides, it is proposed that, even though consumers do not have adequate knowledge of the meaning of third-party certified labels, the credibility associated with them could perform as a holistic heuristic cue of product superiority [6,52,53], so generating a positive evaluation of all its attributes and justifying its higher price. This argument could explain the halo effect suggested above in relation to the environmental superiority attributed in a biased way to all types of sustainability certifications, as well as the effects of such attribution in the valuation of the quality and price attributes of products. Specifically, a new hypothesis is suggested, as follows:

**Hypothesis** **2 (H2).**
*For uninformed consumers, label credibility has positive effects on environmental sustainability inferences (H2a), quality inferences (H2b) and price inferences (H2c), which apply to both organic and non-organic labels.*


Based on the previous literature [8,16,19,21,38], it is expected that providing consumers information about the meaning and certification process of different types of third-party sustainability labels plays a relevant role in the prevention of the biased assessment of the attributes of sustainable products. This unbiased thinking should be appreciated both in making correct judgments about label credibility and environmental sustainability, quality and price inferences, and in previously hypothesized heuristic relationships between these variables.

On the one hand, informed consumers are expected to have greater knowledge to give more credibility and attribute more ecological properties to organic labels [55,78], also reinforcing their expectations of quality and price. In the case of non-organic labels, the provided information will lead to reducing the inferences of environmental sustainability and credibility to judge the ecological meaning but, as it is still an official certification assigned to superior products, high expectations of quality and price should be maintained. Therefore, a new couple of hypotheses is suggested:

**Hypothesis** **3 (H3).**
*For organic labels, informed consumers show higher environmental sustainability inferences (H3a), label credibility (H3b), quality inferences (H3c) and price inferences (H3d) than uninformed consumers.*


**Hypothesis** **4 (H4).**
*For non-organic labels, informed consumers show lower environmental sustainability inferences (H4a) and label credibility (H4b), and the same quality inferences (H4c) and price inferences (H4d) than uninformed consumers.*


On the other hand, if knowledge allows third-party certified labels to be used as heuristic cues in an effective way [55,77,78], informed consumers should have reasoned evidence to strengthen the link between the credibility and meaning of organic labels and the resulting quality and price inferences. However, verification that non-organic labels are not credible or useful certifications regarding the environmental benefits of the product should reduce the ability of both variables to predict the quality and/or price of the product, since they depend on the information about other dimensions of sustainability. Thus, four additional hypotheses are suggested, as follows:

**Hypothesis** **5 (H5).**
*For organic labels, the effects of environmental sustainability inferences on quality inferences (H5a) and price inferences (H5b) are stronger for informed consumers than for uninformed ones.*


**Hypothesis** **6 (H6).**
*For non-organic labels, the effects of environmental sustainability inferences on quality inferences (H6a) and price inferences (H6b) are stronger for uninformed consumers than for informed ones.*


**Hypothesis** **7 (H7).**
*For organic labels, the effects of label credibility on environmental sustainability inferences (H7a), quality inferences (H7b) and price inferences (H7c) are stronger for informed consumers than for uninformed ones.*


**Hypothesis** **8 (H8).**
*For non-organic labels, the effects of label credibility on environmental sustainability inferences (H8a), quality inferences (H8b) and price inferences (H8c) are stronger for uninformed consumers than for informed ones.*


In any case, the literature on marketing and consumer behavior has repeatedly demonstrated that product quality and price dimensions are strongly related to each other [79,80,81], so consumers are willing to pay a premium price for better-quality products. Likewise, various studies analyzing the quality and price-connected variables in relation to dimensions of sustainability tend to confirm the mediating role of quality in willingness to pay a premium price for organic products [20,45,46,47,49]. Hence, we propose the following hypothesis:

**Hypothesis** **9 (H9).**
*Quality inferences have a positive effect on price inferences, which is true for both informed and uninformed consumers and for both organic and non-organic labels.*


## 4. Materials and Methods

### 4.1. Sample, Product Category and Labels

To test the hypotheses, an experimental between-subject study was conducted in 2019 with a sample of 412 business undergraduates at a single university in Spain. Respondents were informed that their participation in the study was entirely voluntary and anonymous and that, by filling in the survey, they provided their consent to participate.

Using a sample of students was considered appropriate, as younger generations tend to show good environmental awareness and a predisposition to sustainable consumption [29,82,83,84]. Additionally, using students from the same branch of studies ensured that all of them had similar prior knowledge about third-party certified eco-labeling. Participants were selected according to convenience criteria among those who attended certain university classes, considering participation in the purchase of unprocessed fruits in daily life as the inclusion criterion in the study, as this was the product category analyzed in the research. Of the total sample, 223 were women (54.1%) and 189 men (45.9%), aged 18 to 26 years old (*M* = 20.42; *SD* = 2.08).

Within the category of unprocessed fruits, the banana was chosen as a specific reference for the study, since, according to the most current data available, it is not only a fruit commonly consumed in Spain [85], but also the most consumed fruit by young people between 18 and 30 years old [86]. Furthermore, the total banana production in Spain is located in the Canary Islands region and it is marketed through a single brand that represents 70% of the total banana consumed at the national level [87], which allowed controlling the effect of possible non-controlled variables related to brand awareness.

Two official sustainability labels are usually used in the commercialization of bananas from the Canary Islands (Plátano de Canarias). On the one hand, around 5% of Spanish banana production is certified as organic [88], which is identified through the organic production logo of the European Union, as established by current EU legislation [89]. On the other hand, the banana from the Canary Islands is also the only product of this type with the recognition of a European Protected Geographical Indication or PGI [90]. This label is awarded to high-quality products whose production takes place in a defined geographical area that is directly related to que quality of the product and the characteristic aspects of the production process. The PGI label offers consumers reliable information about the quality of the products they purchase at the time that enhances the work of the producers and encourages the maintenance and development of the territories and the local economy [65]. Therefore, and in accordance with the purposes of this research, the hypotheses were tested through two officially certified labels at the European Union level and linked to two different aspects of environmental (EU organic logo) and socio-economic (PGI logo) sustainability for the same product category (Figure 1).

### 4.2. Procedure and Measures

The study participants were distributed in two conditions with a similar composition: information condition (*n* = 202) and non-information condition (*n* = 210). The research procedure used in the information condition was developed in two sessions. In the first one, participants received a 15 min talk on organic production, third-party certification procedures and types of sustainability labels. Two weeks later, in the second session, participants were shown the images of the EU organic logo and the PGI label, then having to answer in each case a questionnaire about the credibility of the labels and environmental sustainability, quality and price inferences. Participants assigned to the non-information condition attended only the second session and responded to the same questionnaire without receiving any prior explanation of the labels.

The questionnaire used in the second session of the study was developed for the purposes of the research, based on an in-depth review of the specialized literature and the judgement of three expert researchers on the subject. Likewise, a pretest was carried out with a small subsample of students, who were consulted about possible difficulties in understanding the content of the scales. Following Podsakoff et al. [91], to minimize social desirability and acquiescence biases, respondents were requested to answer the survey anonymously and as honestly as possible.

To measure environmental sustainability inferences associated with each label, participants were presented with a list of seven basic defining characteristics of an organic product in accordance with current European regulations on organic farming [89]. Then, participants had to indicate the extent to which they associated each label with a banana of the concrete characteristic. According to previous studies [21,35,56], label credibility was measured by asking participants to rate each label in seven indicators of trustworthiness (e.g., official, reliable, objective, etc.). Based on previous studies on the halo effect, we defined banana quality inferences around five sensory and health-related properties [41,51], and participants were asked to rate the extent to which they associated the two labels with each property. Finally, we used a single item to measure the participants’ inference of having to pay a premium price for products labeled with the two logos analyzed [21,41,51]. Five-point Likert-type scales (ranging from “1 = nothing at all” to “5 = completely”) were used for the items of all scales.

### 4.3. Data Analysis

The research model was examined through a Partial Least Squares (PLS) structural equation analysis using the SmartPLS 3.0 (SmartPLS Gmbh, Boenningstedt, Germany) statistical program [92]. Unlike covariance-based methods, PLS aims to maximize the variance explained by indicators and latent variables by estimating ordinary least squares and principal component analysis. In this context, data processing responds to the creation of optimal linear predictive paths with minimal demands on measurement scales, residual distributions, and sample sizes [93,94,95]. Therefore, compared to maximum likelihood methods, the PLS approach is better suited to the requirements of predictive–explanatory applications [96,97,98], as in this research.

For reflective measurement approaches, the PLS procedure allows estimating the research model in two stages [99]. The first one of these stages evaluates the strength of the measurement model by observing the reliability of the items, the internal consistency and the construct validity. The second stage focuses on the estimation of the fit parameters for the structural model and reports on the fulfilment of the research hypotheses. In this second stage, two multigroup PLS analyses were carried out [100] to compare the path coefficients obtained for participants in information and non-information conditions in relation to the EU organic logo and the PGI label (H1, H2, H5, H6, H7, H8 and H9). A bootstrapping procedure with 500 resamples was applied to determine the statistical significance of each estimated path. The differences in path coefficients were tested based on the resampling estimates for the standard errors of the structural paths gained from bootstrapping [101]. We also performed two multivariate analyses of variance (MANOVA) with IBM SPSS Statistics 26, to check for differences in research variables between informed and uninformed participants for the two labels analyzed (H3 and H4).

## 5. Results

### 5.1. Measurement Model

The measurement model is shown in Table 1. Item reliabilities (*λ*) were above the threshold of 0.50 [93,96], according to a significance level of *p* < 0.05 and calculated on the basis of 500 bootstrapping runs. 

The internal consistency of the scales was also ensured, since Cronbach’s alpha (*α*) and composite reliability (*ρ_c_*) indices were all above the critical threshold of 0.70 [96,102,103]. Convergent validity was examined using the values of the Average Variance Extracted (AVE), which were above the minimum benchmark of 0.50 [93,102].

To test the discriminant validity, it was verified that the manifest variables correlated more strongly with their associated latent variable than with any other latent variable [96]. As Table 2 shows, the square roots of the AVE values (diagonal elements) were larger than the standardized correlations between constructs (off-diagonal elements), thus suggesting satisfactory discriminant validity [102]. The research model also achieved a Standardized Root Mean Square Residual (SRMR) of 0.054 and a Normed Fit Index (NFI) value of 0.904, which are considered appropriate for PLS methods [104].

As all the measures of the study variables were obtained from the same source, a full collinearity test based on Variance Inflation Factors (VIF) was conducted [105] to determine common method bias based on a common method factor [91]. VIF values were below the critical threshold of 3.3, thus leading to the conclusion that common method bias was not a major concern in this study.

### 5.2. Multigroup Comparison

Once the reliability and the validity of the measurement model were tested, PLS was used to evaluate the structural model separately in both information and non-information conditions for each label (Table 3, Figure 2 and Figure 3).

According to H1a, environmental sustainability inferences were significantly related to quality inferences for uninformed participants, both in the case of organic (*β* = 0.67, *p* < 0.001) and PGI labels (*β* = 0.54, *p* < 0.001). However, H1b was rejected, as the environmental sustainability inferences had no effect on the price inferences.

Regarding H2, the credibility of the label had significant positive effects on environmental sustainability and price inferences under the non-information condition, which was maintained for both the EU organic label (*β* = 0.72, *p* < 0.001; *β* = 0.26, *p* < 0.01) and the PGI label (*β* = 0.54, *p* < 0.001; *β* = 0.17, *p* < 0.05). The effect of label credibility on quality inferences was significant only for the PGI label (*β* = 0.24, *p* < 0.01). These results confirm H2a and H2c and provide partial support to H2b.

Two MANOVA tests were performed to examine how informed and uninformed participants differed in their perceptions of the two labels analyzed (Table 4).

In the first one of these analyses, related to the EU organic logo, the results showed a statistically significant difference between the two groups for the combined dependent variables (*F*_(4, 394)_ = 8.10, *p* < 0.001, Wilks’ Lambda: 0.92). Then, differences were tested separately for each dependent variable, considering a Bonferroni-adjusted alpha level of *p* < 0.0125. As expected, participants in the information condition showed higher environmental sustainability inferences (*F*_(1, 397)_ = 17.81, *M* = 3.58 > *M* = 3.19), label credibility (*F*_(1, 397)_ = 24.19, *M* = 3.72 > *M* = 3.22) and price inferences (*F*_(1, 397)_ = 7.60, *M* = 3.48 > *M* = 3.16). Therefore, H3a, H3b and H3d were supported. However, H3c was rejected because there were no differences in quality inferences between the two groups.

In the second MANOVA analysis, carried out for the PGI label, the difference between the information and non-information groups for the combined dependent variables was not statistically significant. The differences between the groups were also not significant for the variables of environmental sustainability inferences, label credibility and price inferences, although it was found that the uninformed participants showed higher quality inferences than the informed participants (*F*_(1, 404)_ = 7.03, *p* < 0.0125, *M* = 3.10 > *M* = 2.85). These results reject H4a, H4b and H4c, and support H4d.

Regarding the moderating role of information in the relationships between variables, the positive effects of environmental sustainability inferences on quality inferences remained significant in the information condition, both for the EU organic label (*β* = 0.54, *p* < 0.001) and for the PGI label (*β* = 0.63, *p* < 0.001), even when the differences between the path coefficients obtained in the non-information and information conditions were not statistically significant. Likewise, the effects of the environmental sustainability inferences on price inferences remained non-significant in the information condition for the two labels. These results reject H5a, H5b, H6a and H6b.

As expected in H7a, the positive effect of the credibility of the EU organic label on environmental sustainability inferences increased significantly for the information condition (*β* = 0.82, *p* < 0.001; *t* = 20.06, *p* < 0.05). However, the positive effect on price inferences remained significant (*β* = 0.26, *p* < 0.01) without differences in the multigroup comparison, and the effects on quality inferences were non-significant under the two conditions, thus rejecting H7b and H7c.

For the PGI label, credibility had significant positive effects on environmental sustainability inferences also in the information condition (*β* = 0.56, *p* < 0.001), while the effects of label credibility on quality and price inferences turned into non-significant. However, according to the multigroup comparison, the differences between the path coefficients obtained for label credibility under the two conditions were not significant. These results reject H8a, H8b and H8c.

Finally, in support of H9, the positive effect of quality inferences on price inferences was significant, both for uninformed and informed participants, and both for the EU organic label (non-information condition: *β* = 0.26, *p* < 0.001; information condition: *β* = 0.31, *p* < 0.001) and the PGI label (non-information condition: *β* = 0.40, *p* < 0.001; information condition: *β* = 0.41, *p* < 0.001).

In the context of the above results, and regarding the EU organic logo, the model explained 53% and 33%, respectively, of the variance of quality inferences and price inferences in the case of uninformed participants, and 40% of the variance of both variables in the case of informed participants. Regarding the PGI logo, the model explained 48% of the variance of quality inferences and 27% of the variance of price inferences under the non-information condition, while it explained 47% of the variance of quality inferences and 24% for price inferences under the information condition (see Table 3, and Figure 2 and Figure 3).

## 6. Discussion

The obtained results show that heuristic thinking is a fairly frequent phenomenon in the interpretation of third-party certified sustainability labels, which can lead to a biased assessment of the product when consumers do not properly interpret their meaning and legitimacy. In this sense, one of the main contributions of the study to the previous literature is that the halo effect that leads to infer a superior quality of the product when it is identified as organic [38,43,50,51] can be generated by different types of sustainability labels, regardless of their meaning. Thus, this paper supports the idea that the frequent ignorance of concepts related to the environmental superiority of products [18,30,31,34,35,38] could lead to infer ecological properties and a higher global quality in any product certified as sustainable by a third party [32,33], due to an inferential process of interconnections between different attributes of sustainability and quality, that work even when no information about them is available [14].

As a possible explanation for this phenomenon, the results also suggest that the environmental superiority of the product is inferred from the credibility given to the certifying source to support such judgment. Once again, it is noteworthy that this relationship has been found both for the EU organic logo and the PGI label, the latter having no ecological significance. Thus, consumers seem to compensate for their lack of knowledge about the meaning of the labels with the trust they generate [6,18,21,68,69,70] by carrying out a biased assessment of the ecological attributes of the product when the certification does not have scope to legitimize them [52,53].

However, contrary to expectations, the credibility of the label was only associated with quality inferences by uninformed participants regarding the PGI label, but not in the case of the EU organic logo. Thus, it seems that when the label has a more obvious green connotation, credibility is not automatically connected to the inferences on a higher quality of the product, but rather these derive directly from the environmental superiority judgment itself. Nevertheless, the premise that the source credibility is enough to generate biased positive expectations at the level of all product attributes was met with the non-organic sustainability label. These results tinge previous arguments about the desirability of raising consumer trust in third-party certification labels [18,21,22], pointing out the need for this trust to be based on knowledge substantiated on the meaning of such labels, so that the consumer can attribute them credibility adjusted to the scope of their legitimacy.

Together with the above, and unlike previous studies [21,41,48,51], this research did not find a relationship between inferences of environmental sustainability and inferences of product price. Rather, the justification for the price of the products identified by the EU organic and PGI labels was the result of quality inferences and the credibility attributed to the label. Therefore, it seems possible to conclude that credibility does not always act as a quality heuristic cue, but as a cue to justify the premium price of sustainable products. Thus, the quality and price inferences of the products identified by the sustainability tags are constructed, according to the obtained results, through different heuristic tracks provided by the labels, so that the ecological sustainability inferences improve the perceived quality and the justification for the premium price derives from such expectations on quality along with the credibility of the certifying source. In other words, consumers seem to be aware of the existing link between environmental sustainability and the higher quality of food products, but not of the higher production costs of organic farming, then intuitively justifying the higher price of such products, based on the credibility of the label. In short, it appears that consumers may be willing to pay a premium price for certified products, even when they are unaware of the attribute of sustainability that justifies this higher price.

In the context of the conclusions presented so far, another key finding of this study was that providing accurate information on the sustainability certification processes and the meaning of different types of labels did not have a significant enough impact on the research variables and the relationships between them to avoid biased heuristic thinking in product evaluation. Thus, the potential cognitive biases identified in previous paragraphs regarding the interpretation of attributes of environmental superiority, quality and price from the official sustainability labels are extensible, in general terms, to informed and uninformed consumers. Although it was found that information reinforces the perception of credibility and attribution of an adequate meaning to the European eco-label, as well as the positive link between both variables, it was not possible to obtain correct judgments from the informed participants about the credibility and meaning of the PGI label. Likewise, although some of the results from the intergroup comparisons carried out on the relationship between label credibility and environmental sustainability, quality and price inferences were in line with expectations, no statistical support was obtained for the research hypotheses suggesting the information to the consumer as an effective tool to correct the biased perception of certified sustainability labels [55,77,78]. In line with these results, some authors argue that providing information has rather short-lived effects on pro-environmental behavior and sustainable consumption in the medium and long term [106,107,108,109], so a combination of the information with other complementary tactics is necessary [110,111,112,113]. In this sense, given that the time that elapsed between the reception of the information by the participants in the study and the experimental phase when they had to interpret the attributes of the product was only two weeks, it can be said that the intervention did not allow the stimulation of any type of rational thinking oriented to an adequate interpretation of the labels, so being ineffective in neutralizing biased heuristic thinking.

At any rate, the general conclusion that the halo effect and source credibility bias are automatic and information-resistant processes that can lead to purchasing decisions based on unfounded inferences about product properties has important practical implications. Specifically, it seems necessary to work so that third-party sustainability certifications serve as more effective heuristic cues for consumers, allowing them to reach quick conclusions about product attributes with little cognitive effort when making purchasing decisions [4,114]. In other words, sustainability labels and seals should be presented as simple indicators that explicitly identify in which dimensions a product is sustainable and, equally important, in which dimensions it is not [32]. For example, linking certifications to higher-level information tags based on sustainability traffic light labeling—or other related systems—that, using colors or other easily understood visual symbols, clearly indicate the extent to which a product is triple environmentally, socially and economically responsible, could contribute to effective heuristic thinking in promoting responsible consumption, as well as to prevent biased interpretations and, consequently, loss of confidence in sustainability certifications.

Furthermore, enhancing the ability of consumers to attribute appropriate credibility and meaning to different sustainability labels should be accompanied by encouraging them to carry out justified heuristic evaluations of product quality and price. This purpose necessarily involves the tasks of clarifying the relationship between the different levels and aspects of sustainable production and the different dimensions of the final quality of the product, as well as of justifying the cases in which such final result should or should not translate into a higher price for the consumer. All of this can only be achieved if citizen awareness and training measures in the culture of sustainable consumption, implementing in them a solid basis of knowledge that, once internalized, provide cognitive capacity enough to make accurate heuristic inferences in actual purchase situations [10,12,55,76,77,78]. In this sense, although this research has shown that simple interventions based on providing one-time information about the certification process and the meaning of different labels are not effective in favoring the appropriate evaluation of the product attributes, the value of continuing education for consumers on issues of sustainable behavior and consumption should not be overlooked. What is more, there is ample evidence that one is unlikely to engage in more deliberated forms of sustainable consumption if one is not informed about the problem, potential positive actions, and consequences [109,112,115]. From this view, it seems appropriate to address a greater and better awareness of consumers in terms of sustainability, so allowing the achievement of a greater culture of responsible consumption and leading to a greater individual interest in the search for reliable information on the attributes of sustainability, while giving the individuals strong argument about the implications of sustainable production for the quality and price of the products.

Nonetheless, this research suffers from certain limitations that should be addressed in future research. First, it should be noted that, although the research carried out has allowed us to reach some interesting conclusions and that the percentages of variance explained in the dependent variables of the model were quite substantial in the different conditions of the study, several research hypotheses that were initially raised could not be confirmed. In this sense, and even though such a pattern of results has significant theoretical and practical implications in the context of previous literature, it is necessary to reinforce its validity through other studies that replicate the research and try to introduce improvements in terms of the experimental design, the measurement of variables, and the consideration of additional constructs.

Second, the research was conducted with a sample of students from the same branch of studies and in a single Spanish university. In this sense, and although this group was considered suitable for the purposes of the research due to the verified greater environmental awareness and predisposition to sustainable consumption of the young generations [29,82,83,84], it is no less true that the generalization of the obtained results is compromised. Therefore, new studies are required to test the research model in other consumer samples with greater sociodemographic diversity. Likewise, it seems interesting to make comparisons with consumers from other countries, considering cultural variables that can affect the knowledge and awareness of consumers and that can moderate many of the relationships that were hypothesized in the explanation of the biased inferences about the product derived from the interpretation of sustainability labels.

Third, the research was focused on a single product category, marketed under a single guaranteed brand that was supposed to be well known to the study participants. Thus, the credibility of the labels considered, and the inferences of environmental sustainability, quality and price could be influenced, at least partially or to some extent, by the confidence of participants in the guaranteed brand. In this sense, it seems interesting that future research considers brands of diverse notoriety and different congruence with sustainability certifications, to identify possible factors involved in heuristic thinking and incurrence of cognitive biases. Similarly, it is suggested to test the validity of the model in other categories of food products, as well as to test it in relation to the purchase of durable consumer products, where the purchase decision process could lead to a more rational analysis of the product attributes and to moderate the relationships between the inferences about sustainability, quality and price.

Fourth, the study was focused on analyzing the credibility and the meaning of environmental sustainability associated with a third-party certified organic label (EU organic logo) and a socio-economic sustainability certification (PGI label), both related to the same product category. However, the credibility and meaning of both labels were not studied in relation to the dimensions of social and economic sustainability, which could introduce some nuances in the prediction of quality and price inferences. New attempts could address this research opportunity and also consider other types of third-party certified sustainability labels as, for example, fair trade labels [32,33], thus offering a more complete picture of the heuristic interconnections that are established between different dimensions of sustainability as well as identifying other cognitive biases that may need to be prevented.

Fifth, and related to the above, it would be interesting to analyze the working of the halo effect and source credibility bias in the interpretation of certain ambiguous commercial claims not certified by third parties and that frequently appear on food product packaging to suggest a greater environmental superiority of the product. As, according to various studies, consumers face difficulties in correctly interpreting such claims and differentiating them from officially certified sustainability labels [8,53,68], it seems appropriate to study to what extent the heuristic inferences of environmental sustainability, quality and price reinforce the greenwashing actions of many companies, since consumers do not have the adequate knowledge to base their judgments.

Finally, it should be noted that this research has allowed an explanation to the assessment of the product by consumers when they are exposed to certified sustainability labels, but the conclusions provided do not have scope enough to predict how the inferences of the product attributes are transferred to real acts of purchase. In this sense, new studies should consider additional behavioral variables and be based on research contexts closer to real purchase situations.

## Figures and Tables

**Figure 1 foods-10-02512-f001:**
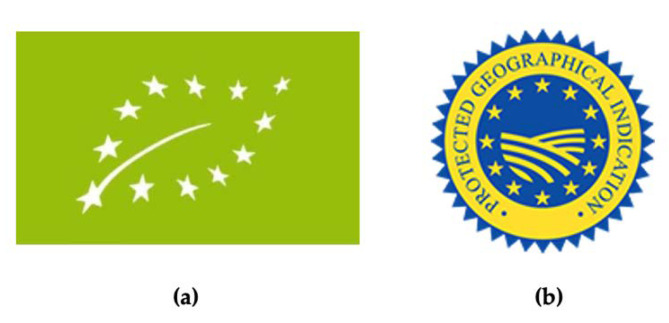
Third-party certified labels used in the research. (**a**) EU organic label (environmental sustainability). Regulation EU 2018/848 of the European Parliament and of the Council of 30 May 2018; (**b**) PGI label (socioeconomic sustainability). Regulation EU 1151/2012 of the European Parliament and of the Council of 21 November 2012.

**Figure 2 foods-10-02512-f002:**
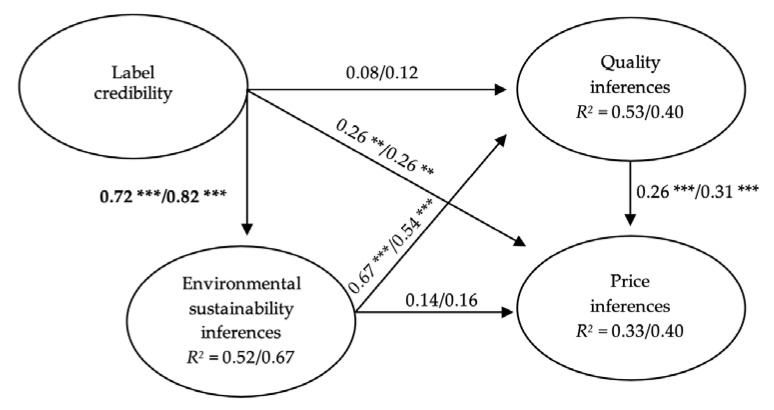
Structural model for the EU organic label (non-information/information). Note. Significant differences are in bold. ** *p* < 0.01; *** *p* < 0.001 (based on *t*
_(499)_, two-tailed test).

**Figure 3 foods-10-02512-f003:**
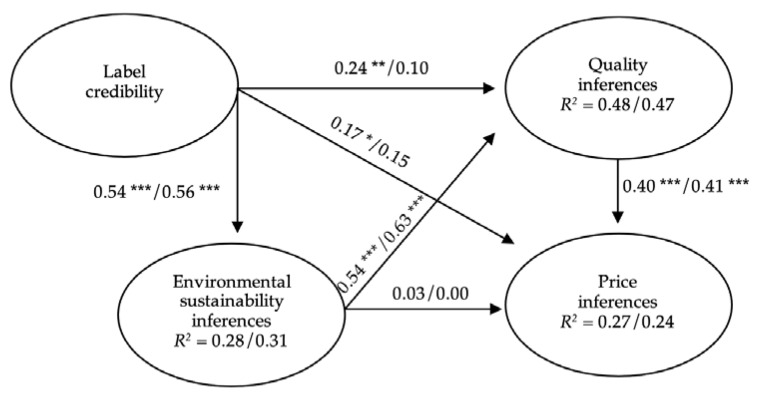
Structural model for the PGI label (non-information/information). Note. * *p* < 0.05; ** *p* < 0.01; *** *p* < 0.001 (based on *t*
_(499)_, two-tailed test).

**Table 1 foods-10-02512-t001:** Measurement model assessment.

Variables and Items	Loadings
*Environmental sustainability inferences* (*α* = 0.91; *ρ_c_* = 0.93; AVE = 0.65)	
Obtained without the use of chemical pesticides	0.85 ***
Obtained without the use of chemical fertilizers	0.83 ***
Obtained without artificial additives	0.82 ***
Not genetically modified	0.84 ***
Produced in an environmentally friendly way	0.80 ***
Obtained respecting the natural growth rate of the plant	0.81 ***
Obtained with cultivation methods adapted to the optimal use of local conditions	0.68 ***
*Label credibility* (*α* = 0.93; *ρ_c_* = 0.94; AVE = 0.70)	
Credible	0.86 ***
Objective	0.83 ***
Compelling	0.85 ***
Reliable	0.87 ***
Official	0.79 ***
Useful	0.84 ***
Relevant	0.86 ***
*Quality inferences* (*α* = 0.82; *ρ_c_* = 0.87; AVE = 0.58)	
Standing out due to its flavor	0.78 ***
With a better appearance than others	0.72 ***
With a greater durability than others	0.72 ***
Beneficial for health	0.79 ***
With noticeable nutritional properties	0.80 ***
*Price inferences*	
With a higher price than others	--

Note. *α* (Cronbach’s alpha); *ρ_c_* (composite reliability); AVE (average variance extracted) *** *p* < 0.001 (based on *t*
_(499)_, two-tailed test).

**Table 2 foods-10-02512-t002:** Correlation matrix and square roots of AVE.

	1	2	3
1. Environmental sustainability inferences	0.81		
2. Label credibility	0.69 **	0.84	
3. Quality inferences	0.65 **	0.52 **	0.76
4. Price inferences	0.47 **	0.47 **	0.52 **

Note. Square roots of AVE (average variance extracted) are in parentheses. ** *p* < 0.01.

**Table 3 foods-10-02512-t003:** Multigroup comparison.

	EU Organic Label			PGI Label		
Paths	Non-Information *β* (*t*)	Information *β* (*t*)	Differences *β*-*β* (*t*)	Non-Information *β* (*t*)	Information *β* (*t*)	Differences *β*-*β* (*t*)
ESI → QI	0.67 *** (91.65)	0.54 *** (65.02)	0.24 (11.58)	0.54 *** (64.80)	0.63 *** (88.70)	0.09 (0.79)
ESI → PI	0.14 (13.88)	0.16 (15.09)	0.92 (0.11)	0.03 (0.28)	0.00 (0.03)	−0.03 (0.18)
LC → ESI	0.72 *** (182.25)	0.82 *** (297.90)	0.10* (20.06)	0.54 ***(81.99)	0.56 *** (83.19)	0.03 (0.31)
LC → QI	0.08 (11.24)	0.12 (14.12)	0.03 (0.28)	0.24 ** (30.15)	0.10 (13.35)	−0.14 (12.17)
LC → PI	0.26 ** (29.15)	0.26 ** (26.27)	0.00 (0.04)	0.17 * (20.37)	0.15 (16.66)	0.03 (0.22)
QI → PI	0.26 *** (34.69)	0.31 *** (37.13)	0.05 (0.49)	0.40 *** (43.02)	0.41 *** (43.79)	0.02 (0.17)
*R*^2^ ESI	0.52	0.67		0.28	0.31	
*R*^2^ QI	0.53	0.40		0.48	0.47	
*R*^2^ PI	0.33	0.40		0.27	0.24	

Note. ESI (environmental sustainability inferences); LC (label credibility); QI (quality inferences); PI (price inferences). * *p* < 0.05; ** *p* < 0.01; *** *p* < 0.001 (based on *t* _(499)_, two-tailed test).

**Table 4 foods-10-02512-t004:** Differences between information and non-information conditions.

		EU Organic Label			PGI Label		
	Condition	Mean	SD	*F* _(1, 397)_	Mean	SD	*F* _(1, 404)_
Environmental sustainability inferences	Non-information	3.19	0.90	17.81 *	3.12	0.83	5.73
	Information	3.58	0.97		2.92	0.88	
Label credibility	Non-information	3.22	0.97	24.19 *	3.39	0.85	4.31
	Information	3.72	1.08		3.21	0.94	
Quality inferences	Non-information	2.90	0.83	0.42	3.10	0.78	7.03 *
	Information	2.97	0.86		2.85	0.79	
Price inferences	Non-information	3.16	1.12	7.60 *	3.42	1.06	1.66
	Information	3.48	1.15		3.28	1.09	

Note. EU organic label: F_(4, 394)_ = 8.10, *p* < 0.001, Wilks’ Lambda: 0.92; PGI label: F_(4, 401)_ = 2.03 (non-significant), Wilks’ Lambda: 0.98, * *p* < 0.0125 (Bonferroni adjusted alpha level).

## Data Availability

Data available is available on request from the corresponding author.

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
