# Peer review of "Halo Effect and Source Credibility in the Evaluation of Food Products Identified by Third-Party Certified Eco-Labels: Can Information Prevent Biased Inferences?"

_foods, 2021, doi:10.3390/foods10112512_

Round 1
Reviewer 1 Report
Dear Authors,
thank you for this interesting manuscript. The manuscript is interesting, but some minor revision are necessary.
The literature review can be improved, please make a more critical analysis (now it is too much descriptive) and move your "hypothesis" to a separate section.
by my point of view, minor revision are requested with reference to section 2 “literature review".
I suggest to improve the it. I propose to separate all the “hypothesis” indicated in the literature section in a new, separate section of the manuscript: in my opinion they do not have to be mentioned in the literature section, as the literature review has to be related to the background of the study and not to the objective/the aim of the research. The literature section is moreover too much descriptive and does not present a critical analysis of the topic.
Please try to better highlight the novelty of your research.
Author Response
We greatly appreciate the kind comments of Reviewer 1, as they have been very helpful in improving the paper. Certainly, our general impression with the first version of the manuscript was to be conducting a critical analysis on the previous literature on the halo effect generated by official eco-labels, defending that: 1) this effect is much deeper than is thought and can be extensible to all types of sustainability labels (section 2.1); 2) the halo effect and its extension to all types of sustainability labels can be explained from a previous source credibility bias (section 2.2); and 3) providing the consumer with reliable information on the meaning and legitimacy of sustainability labels should help prevent such cognitive biases (section 2.3). These three axes are also described in the Introduction section.
After reading the reviewer's comments, we realized that the critical approach was not clear, so we have rewritten the sections of the Literature Review. We trust that the critical analysis we want to raise is better appreciated now. In any case, we have kept some sentences and paragraphs with a more descriptive style, since it seems important to us that the article can be easily understood by a wide audience, which may not be familiar with the subject of the research. Likewise, we hope that the effort to synthetically describe the results of different relevant previous studies will be appreciated.
On the other hand, the hypotheses were initially raised within the literature review subsections as this is the structure used in recent articles published in the journal and based on the same PLS methodological approach (see, for example, Bîlbîie, A. ; Druic ˘a, E .; Dumitrescu, R .; Aducovschi, D .; Sakizlian, R .; Sakizlian, M. Determinants of Fast-Food Consumption in Romania: An Application of the Theory of Planned Behavior. Foods 2021, 10, 1877). Following the revieewer’s suggestion, we have moved the hypotheses to a separate section. Consequently, we have renumbered the rest of the sections.
We trust that both changes will clarify the novelty and contribution of the research.
Reviewer 2 Report
This is a very well conducted study and very well written paper. I only have some minor comments as follows:
Line 289: what do you mean by “strange variables”?
Line 487: “f14”?
Line 512: should it be “a quality heuristic”?
Line 576: Should it be “not effective”?
Line 599: Better with “suitable for the purpose”?
A general comment is that, even though the paper is very well written, some sentences are long with many commas (no response needed). Long sentences and long words do reduce the readability of papers and may actually reduce the impact of the research (Sawyer, Laran, and Xu 2008).
Author Response
We are glad that the reviewer found the article interesting and we welcome his/her comments, which have helped us to improve the understanding of the article. Specifically, the following corrections have been made:
- Line 315 (prior line 289). In an experiment, “strange variables” are non-controlled variables that can affect the independent variable. In this case, as we focus in a single and well-known Spanish brand of bananas, we control the potential effects of those variables related to brand awareness. As it can be confusing, we have replaced the term by “non-controlled variables”.
- Line 514 (prior line 487). We have corrected the typographical mistake (“f14or” is “for”).
- Line 539 (prior line 512). We have corrected the sentence (“a quality heuristic”).
- Line 603 (prior line 576). We have corrected the sentence (“are not effective”).
- Line 626 (prior line 599). We have corrected the sentence (“suitable for the purpose”).
We also appreciate the reviewer's comment regarding the use of long sentences in the text and will consider the results of Sawyer et al. (2008) in the future. Certainly, the use of this type of writing is common in our native language and, as it is an article translated into English, the same style has been preserved. Since the reviewer points out that it is not necessary to respond to his/her comment, no changes have been made to the text. However, if the reviewer or the editor finds that a section or paragraph is particularly difficult to understand, we can tackle the task of rephrasing it with shorter sentences.